# Effect of Different Control Strategies on the Heat Transfer Mechanism of Helical Energy Piles

**Pan Wei, Kongqing Li \***[ID]**, Chenfeng Yu and Qiaoyun Han**

School of Civil Engineering, Hunan University of Science and Technology, Xiangtan 411100, China;
21010201029@mail.hnust.edu.cn (P.W.); 22020201101@mail.hnust.edu.cn (C.Y.); lyxc43@163.com (Q.H.)
* Correspondence: likongqing@hnust.edu.cn

**Abstract:** In this paper, numerical simulations of a special energy pile, which constitutes a spiral-injected pipe and one straight discharge pile for Geothermal Heat Pump Systems (SGHEs-P(parallel)), were conducted by Fluent software. The effects of the spiral pitches on the heat transfer rate based on the G-function method and peripheral soil temperature of the pile were investigated under continuous and intermittent operation strategies. The impact of spiral tube sizing on the surface heat transfer coefficients was studied. The results indicated that SGHEs-P may be preferred for office buildings under intermittent operation conditions. For a short period, the temperature profiles and heat transfer efficiency of SGHEs-P were mainly influenced by the fluid type, length of the spiral tube, and spiral pitch. The smaller the spiral pitch, the more uniform the temperature distribution, and the better the heat transfer effect, but the heat transfer per unit depth of pile decreased. The average temperature variation curve of the soil around the energy pile with different spiral pitches was simulated and obtained over time. Meanwhile, the impact of spiral radius, spiral pitch, and spiral tube radius on the convective heat transfer coefficient was also presented. Through data fitting, the formulas for the correction coefficients of spiral radius, spiral pitch, and spiral tube radius on convective heat transfer coefficient were obtained, respectively.

**Keywords:** helical energy piles; intermittent operation; numerical modeling; soil thermal damage; correction factor

## 1. Introduction

Building energy consumption has now been reduced from 40% of global total energy to 36% [1,2]. In most regions of China, refrigeration and heating energy consumption still accounts for 50–70% of building operating energy consumption [3]. To reduce greenhouse gas emissions, cooling and heating energy consumption in buildings should shift from non-renewable to renewable energy. Geothermal energy first entered the public [4] eye as an energy source in 1904. Compared with other renewable energy technologies, ground source heat pump systems (GSHPs) have gradually become one of the most promising renewable energy technologies in the modern architecture field owing to their excellent sustainability, efficiency, and low emissions.

In recent years, GSHPs have been increasingly applied in engineering. GSHPs consist of above-ground heat pump systems and underground heat exchanger systems (GHEs). There are many types of GHEs. According to the different forms of buried pipes, common GHE forms include U-type [5], W-type [6], triple U-type [7,8], spiral-type [9,10], etc. Compared with U-shaped energy piles, the burial depths of spiral ground heat exchanger systems (SGHEs) are drastically reduced, making construction less difficult. In 2015, Bezyan [9] and Miyara et al. [11] compared the heat transfer rates and temperature distributions of U-type energy piles, W-type energy piles, and SGHEs with three different gradients using Fluent 2019 software and found that better heat transfer efficiencies were achieved with SGHEs, regardless of whether the heat transfer fluid in the tubes was a

liquid or a gas. In the same year, Park et al. [10,12] and Zhao et al. [13,14] studied the heat transfer characteristics of SGHEs with varying spiral pitches, respectively, through on-site construction, COMSOL calculation software, and the G-function approach. They found that the changes in fluid temperature and internal thermal resistance both decreased as the spiral radius decreased and that heat transfer efficiency was not directly proportional to the length of the internal pipe, suggesting that there might be thermal interference between the adjacent buried pipes in SGHEs. Liu [15] and Cecinato et al. [16] used experimental methods and finite element numerical modeling to investigate the fluid thermal resistance and heat transfer efficiency of SGHEs at different flow rates under different helical shapes and found that the flow rate did not affect the heat transfer efficiency of SGHEs under turbulent conditions. Javadi et al. [17,18] investigated the effect of the spatial arrangement of helical tubes in a pile body on heat transfer efficiency using CFD and found that increasing the number of helical tube arrangements increased heat transfer efficiency while decreasing the number of arrangements led to a more stable heat transfer efficiency. Yang et al. [19] studied the heat transfer process of double-spiral-type energy piles in series by spiral injection and spiral discharge (SGHEs-S) and compared the heat transfer efficiency of six pitches and found that the heat transfer efficiency of SGHEs-S could be improved by shortening the pitch. In their study on heat transfer in energy piles, Yan et al. [20] found that the greater the difference in thermophysical properties between the soil and the pile foundation, the more the accuracy of the heat transfer prediction model is affected. Zhang [21] and Molina-Giraldo et al. [22] studied the effect of simultaneously considering permafrost in the soil, groundwater seepage, and heat transfer through the water pipe of an energy pile and found that increasing the distance between the energy piles reduced the heat transfer effect of the piles but improved year-round performance. Antelmi [23] and Jia et al. [24] investigated the effects of groundwater seepage on heat exchange in energy piles and found that flowing groundwater was conducive to soil temperature recovery and the heat exchange of an energy pile.

To summarize, although helical pitch is an important factor affecting the heat transfer efficiency of SGHEs and can be improved by reducing the pitch, recent studies have found it difficult to quantitatively describe the effects of pitch size on the heat transfer of energy piles, which leads to a limitation in the scope of application. At the same time, energy piles are often buried at shallow depths and only exchange heat with the soil near the surface, which can lead to abnormal changes in soil temperatures and, thus, thermal damage, hindering the extraction of geothermal energy. Although groundwater seepage can effectively curb the problem of thermal damage in the ground, steady seepage is prevalent in layers with a vertical depth of more than 30 m. Surfaces with a vertical depth of less than 30 m are usually affected only by rainwater infiltration, which is limited by unpredictable local weather conditions. Energy piles are typically used in engineering to stabilize soil layers, avoiding unstable soil layers. Therefore, the problem of heat damage caused by shallow energy piles at depths of less than 30 m and the quantitative prediction of heat damage need to be solved urgently. At present, energy piles are often used to meet central heating or cooling needs, which often leads to the continuous operation of geothermal heat pump systems, causing the ground temperature to continue to change without recovery, resulting in a decrease in the effectiveness of heat transfer. In office buildings, however, systems only need to operate during working hours. The strategy of intermittent operation gives the ground enough time to recover and utilize surface geothermal energy with more sustainability.

First, this paper models and validates the energy pile. Then, it intends to simulate the heat transfer process of an SGHPs-P energy pile using Fluent software and mainly focus on the most commonly used helix pitches [10,12–18,24–29] in engineering applications on the heat transfer of SGHEs-P and to analyze the temperature distribution of the energy pile using the G-function method. Finally, it analyzes the effects of an operation pattern (continuous and intermittent) on the ground temperature and the influence of helix pitch, helix radius, and diameter of the helical tube on the helical tube wall surface convective heat transfer coefficient.

## 2. Methods

### 2.1. Physical Model

A double spiral energy pile with spiral feed and spiral discharge in parallel (SGHEs-P) consists of 3 parts in a horizontal direction, as shown in Figure 1. It consists of a 6 m × 6 m × 16 m outer soil layer, two sets of heat exchange tubes consisting of vertical circular tubes connected in series by spiral tubes connected to the water above the spiral tubes, and the material of the pile foundation is homogeneous concrete. Only heat exchange takes place between the above-mentioned structures without material exchange. In the vertical direction, the upper 0.4 m is a vertical round pipe partially connected to the lower part of a spiral pipe with a depth of 15.6 m by a right-angled bend with a bend, and the lower part of the pipe is connected to the end of the spiral pipe and an outlet pipe in series by a straight pipe of 0.1 m using two right-angled bends with a bend.

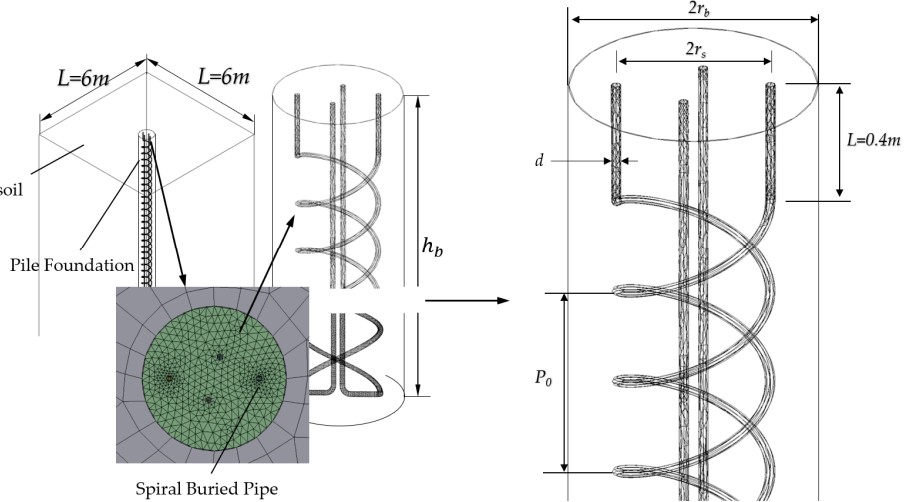

**Figure 1.** Physical model.

### 2.2. Boundary and Initial Conditions

In order to facilitate the comparison of heat transfer performance of different types of energy piles, the computationally relevant thermophysical parameters and design dimensions for the numerical simulations of energy piles and spiral tubes with similar or analogous parameters to those of actual engineering applications of SGHEs and analyses and studies of simulations and analyses in the literature were adopted, as shown in Table 1. The heat transfer fluid in the pipe is water with a velocity of 0.3 m/s.fo; according to the calculation, the Reynolds number (Re) of the water in the spiral pipe is 9316, so the flow state is turbulent model-based simulation of SGHEs-P with Fluent 2019. The material of the soil and the pile foundation is homogeneous and isotropic, with initial temperature of 291.65 K, and the seepage phenomenon is not considered in this paper.

**Table 1.** Design and physical property parameters of energy piles.

| Parametric | Value | Parametric | Value |
|---|---|---|---|
| Inlet fluid temperature/(K) | 308 | Depth/(m) | 16 |
| Inlet flow velocity/(m/s) | 0.3 | Edge length/(m) | 6 |
| Kinematic viscosity/(Pa·s) | $0.805 \times 10^{-6}$ | Helix radius/(m) | 0.2 |
| Pile diameter/(m) | 0.65 | Spiral tube diameter/(mm) | 25 |
| Pr | 5.42 | Helical pitch/P(m) | 0.6 |
| Pile density/(kg/m³) | 2500 | Soil initial temperature/(K) | 291.65 |
| Soil specific heat capacity/(J/kg·K) | 900 | Soil density/(kg/m³) | 1930 |
| Pile thermal conductivity/(W/m·K) | 1.97 | Soil specific heat capacity/(J/kg·K) | 1300 |
| | | Soil thermal conductivity/(W/m·K) | 1.92 |

### 2.3. Comparison of the Effect of Energy Pile Type on Heat Transfer

In the following data processing procedure, the actual spatial coordinates $x$, $y$, $z$, the diameter of the spiral tube ($d$), the radius ($r_s$), and the pitch ($P$) are nondimensionalized using Equations (1) and (2).

$$(X, Y, Z) = (x/d, y/d, z/d) \tag{1}$$

$$SP = P/d, SR = Sd = r_s/d \tag{2}$$

where $X$, $Y$, $Z$ are dimensionless coordinates; $SP$, $SR$, $Sd$ denote the dimensionless dimensions of $P$, $r_s$, and $d$, respectively.

Three types of energy piles were compared: a U-shaped energy pile with diameter of 0.65 m($r_b$ = 0.65 m), an SGHEs-S, and an SGHEs-P (Figure 2) Under the same inlet mass flow, it is found that SGHEs-P-type energy pile has the best heat transfer effect. From 0.5 h to 12 h, the heat transfer of SGHEs-P pile is 120% and 63% higher than that of U-type energy pile and SGHEs-S pile, respectively. From 12 h to 1000 h, the heat transfer of SGHEs-P pile is 45% and 20% higher than that of U-type energy pile and SGHEs-S pile, respectively.

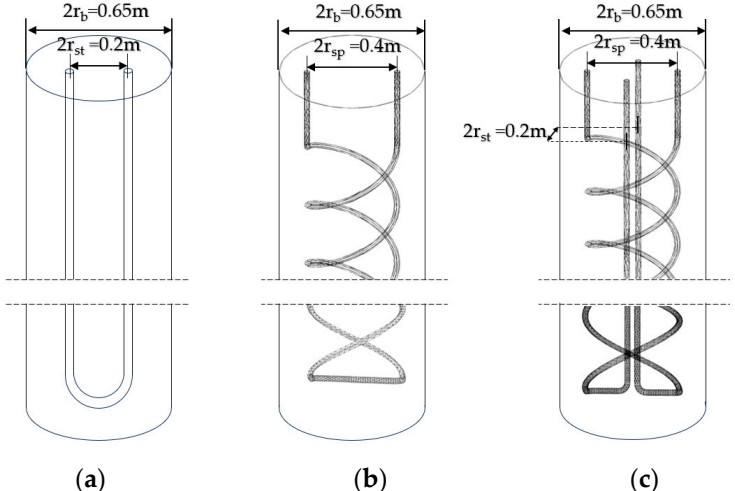

|   |   |   |
|---|---|---|
| (**a**) | (**b**) | (**c**) |

**Figure 2.** Schematic diagram of 3 types of energy piles as: (**a**) U-tube; (**b**) SGHEs-S; (**c**) SGHEs-P.

### 2.4. Grid-Independent Analysis and Model Validation

In energy piles, a structured grid is employed for the majority of the pile; an unstructured grid is utilized at the interfaces between the pile foundation and the earth layer, as well as between the spiral coil and the pile foundation. And the single-turn spiral pipe is encrypted with a structured grid. As shown in Figure 3.

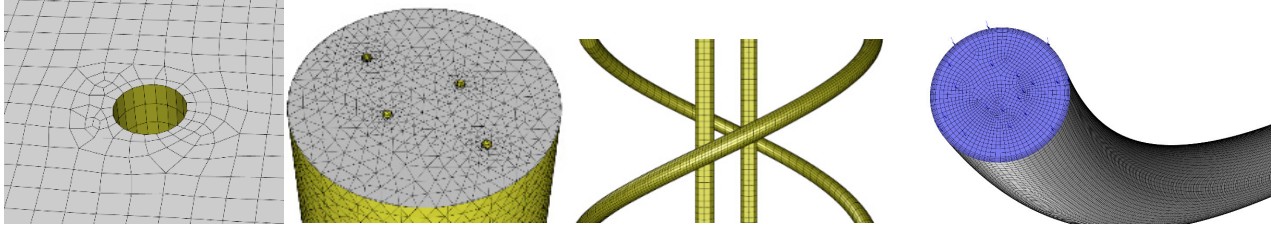

**Figure 3.** Meshing of energy piles and single-loop spiral tube.

In order to obtain grid-independent solutions, we simulated the variation in solutions of a pile and a spiral tube under different numbers of grids, and the error percentage C in the outlet water temperature is defined by Equation (3). The transient water temperature profiles ($T_{out}$) at the outlet of the pile with 2.13 million total grid cells and 2.9 million grids

are shown in Figure 4a. The steady simulation results of a single spiral tube with different mesh numbers are shown in Figure 4b.

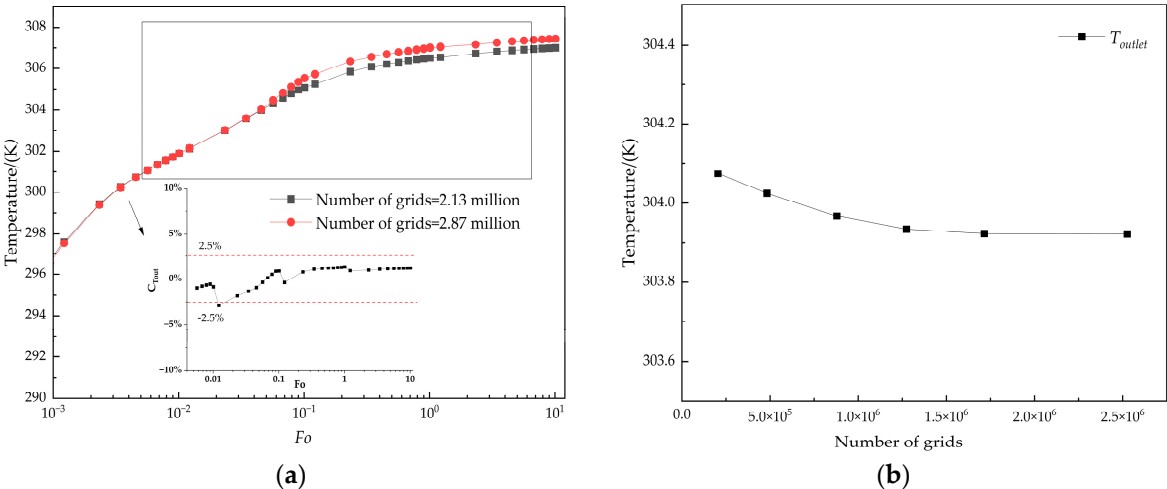

(a)                                                                 (b)

**Figure 4.** Grid-independent analysis as (**a**) energy pile ($T_{out}$), (**b**) single-loop spiral tube ($T_{out}$).

The results showed that the maximum difference in water outlet temperature is less than 3%, and the average difference is less than 1.5% when the total number of grid cells for energy piles increased by 35%. When the total number of grid cells for a single-turn spiral tube is more than 1.25 million, the outlet water temperature had little change.

$$C = (T_{out} - T_{in})/(T_{in} - 273.15) \tag{3}$$

In order to verify the validity of the numerical simulation, the helix energy pile described in reference [30] was simulated. A comparison between the simulated results and the measured data in the literature is shown in Figure 5.

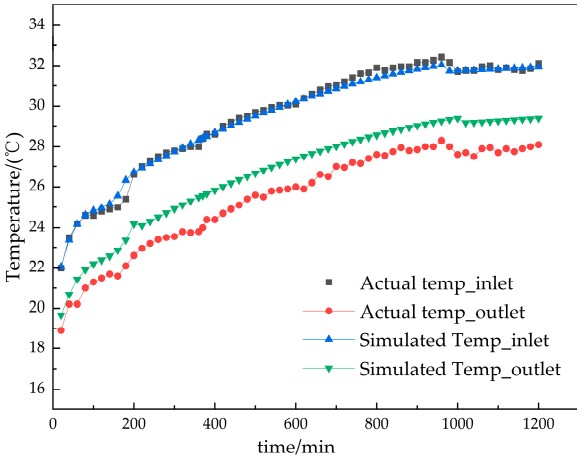

**Figure 5.** Energy pile simulated ($T_{out}$) vs. cited literature ($T_{out}$).

It can be seen that the simulated outlet temperature of water is slightly higher than the measured values. It was mainly caused by seepage around the energy piles, which was not considered in this paper. The outlet temperature relative error between the measured and simulated results after 16 h is below 7%. The simulation results agree with the measured data very well. Therefore, the results of the numerical simulation method used in this paper are credible and meet the requirements of engineering applications.

## 3. Results

*3.1. Influence Law of Spiral Pitch on Heat Transfer Rate and Temperature in Some Local Positions of Energy Piles*

The numerical simulation of an SGHPs-P energy pile with a radius of 0.2 m, a spiral tube diameter of 25 mm, and five kinds of dimensionless spiral pitches (SP = 18, 24, 28, 36, 40) was carried out. The curves for dimensionless temperature, dimensionless time, and dimensionless temperature gradient are defined by Equations (4)–(6).

The dimensionless temperature profiles are displayed in Figure 6 when *Fo* is equal to 1.22 (*t* = 91 h) along the horizontal line where *Y* = 0 and *Z* = 0 in the x-direction and *X* = 0 and *Z* = 0 in the y-direction, and the line is based on the G-function method [14].

$$\theta = k_g h_b (T_i - T_g) / \Phi_f \tag{4}$$

$$Fo = \alpha \tau / r_{sp}^2 \tag{5}$$

$$\partial \theta / \partial R = (\partial T / \partial r_s)(k_g h_b r_s / \Phi_f) \tag{6}$$

where *k* is thermal conductivity; the subscripts *b*, *f*, *g*, *tube*, and *I* stand for the pile wall, fluid, soil, pipe wall, and temperature at a local point on the line, respectively; and *Φ* is heat transfer quantity.

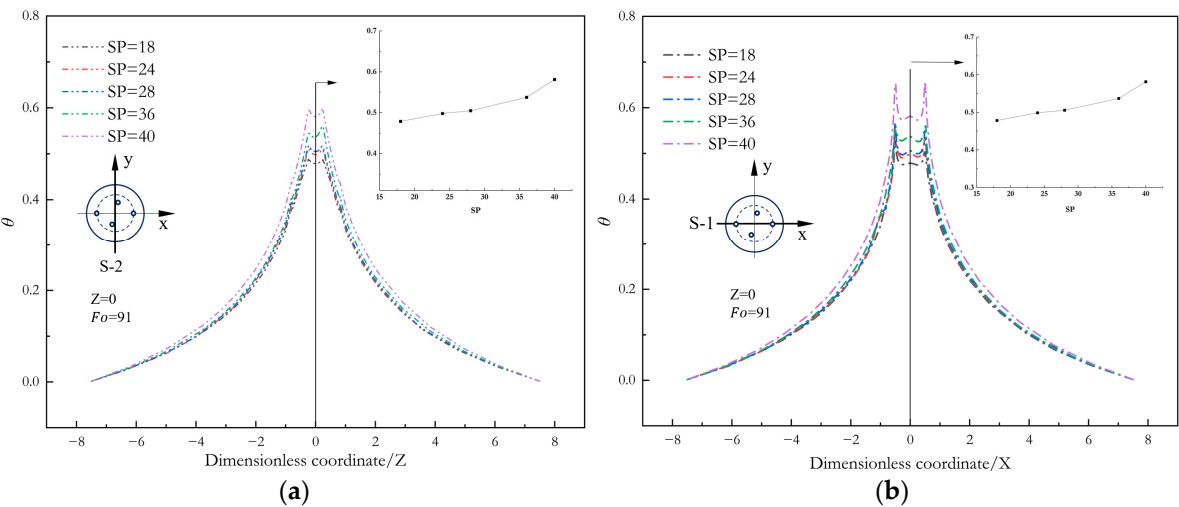

**Figure 6.** Temperature distribution of energy piles as (**a**) horizontal temperature distribution in *x*-axis, (**b**) horizontal temperature distribution in *z*-axis.

Based on the graph shown in Figure 6a, it can be seen that the temperature distribution of SGHPs-P at the *Z* = 0 plane has peaks at *X* = ±0.5. This occurrence may be due to the accumulation of heat. Additionally, some peaks can also be observed at *Y* = ±0.25 in Figure 6b due to the significant impact of the straight outlet tube.

The dimensionless temperature gradient profiles are displayed in Figure 7 when Fo is equal to 1.22 (*t* = 91 h) along the vertical line where *X* = 13 and *Y* = 0, and the line is based on the G-function method [14].

In addition, the temperature differential of the energy piles in the horizontal direction increases while the temperature in its radial direction rapidly drops toward the exterior. As the helical pipe pitch distribution changes, the temperature gradient at the pile base wall in Figure 7 swings frequently, and the magnitude of the fluctuation diminishes as SP decreases.

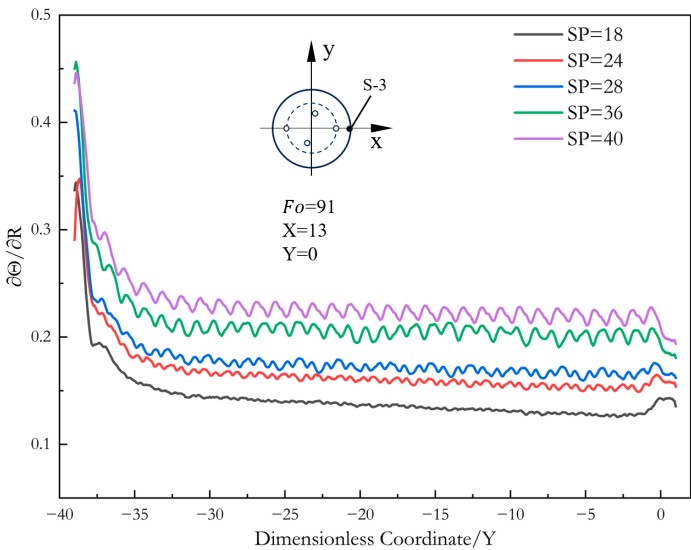

**Figure 7.** Temperature gradient distribution of energy pile walls.

Letting A, B, and C represent three SGHEs-P operating strategies: A represents continuous operation under constant temperature (308 K) at the inlet, B stands for intermittent operation with the same inflow temperature as A, and C indicates intermittent operation in which the inflow temperature data were measured. The changes in cumulative heat transfer ($Q_0$), average pile foundation temperature ($\overline{T}p$), and average soil temperature ($\overline{T}soil$) over time for 1 day in 2 weeks for each of the five *SP*s of SGHEs-P under the same inflow rate are shown in Figure 8. The variations in the energy piles' inlet water temperature over time for each of the three operating strategies are summarized in Table 2.

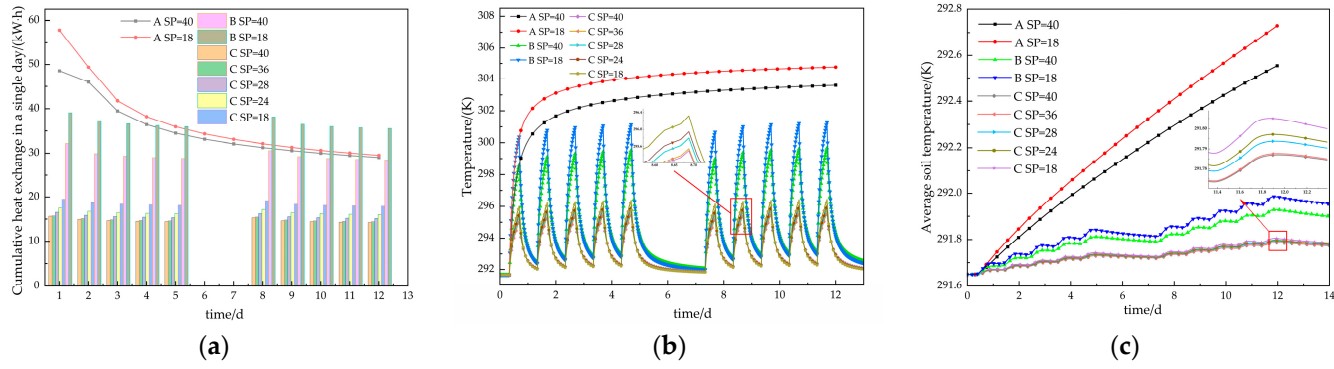

**Figure 8.** Heat exchange and temperature variation in energy piles under three different operating strategies as (**a**) cumulative heat exchange in one day ($Q_0$); (**b**) average temperature of pile foundation ($\overline{T}_p$); (**c**) average soil temperature ($\overline{T}_{soil}$).

**Table 2.** Energy pile inflow water temperature (K) parameters.

| Time | 8:00 | 9:00 | 10:00 | 11:00 | 12:00 | 13:00 | 14:00 | 15:00 | 16:00 | 16:30 |
|---|---|---|---|---|---|---|---|---|---|---|
| Operating strategy A | | | | | 308 (keep all day) | | | | | |
| Operating strategy B | 308 | 308 | 308 | 308 | 308 | 308 | 308 | 308 | 308 | 308 |
| Operating strategy C | 300.03 | 299.64 | 300.13 | 298.98 | 299.13 | 298.95 | 300.42 | 299.52 | 301.12 | 300.91 |

The cumulative heat transfer ($Q_0$) of SGHEs-P for a single day is shown to be uniformly affected by t and SP in Figure 8a, meaning that $Q_0$ declines as *t* and *SP* increase. $Q_{0,SP = 18}$ is greater than $Q_{0,SP = 40}$ by 4.14%, 24.89%, and 25.66%, respectively, when the SGHEs-P are operated using strategies A, B, and C for 5 days. The percentages are 2.13%, 24.99%, and

25.97% greater if they continue until the 10th day. Moreover, $Q_{0,A} < Q_{0,B}$ when $t$ extends for more than 5 days. Figure 8b illustrates how the energy pile foundation's average temperature ($\overline{T}p$) rises quickly with t during continuous operation up until the second day, at which point the temperature progressively stabilizes. This shows that the heat transfer effect substantially diminishes as the pile foundation's heat transfer capacity reaches its limit and the heat transfer between the fluid, pile foundation, and soil approaches the dynamic equilibrium. At the same time, $\overline{T}p$ is greatly impacted by $SP$. $\overline{T}p$ decreases as $SP$ decreases. For instance, on day nine, the temperature rise $\Delta\overline{T}_{p,SP=18}$ for the three operation techniques in a single day is 85.4%, 127.36%, and 127.67% higher than the temperature rise $\Delta\overline{T}_{p,SP=40}$ for the energy piles. Figure 8c illustrates how the increase in soil temperature ($\Delta\overline{T}_{soil}$) increases with $t$ and decreases with $SP$ when using strategies A, B, and C for 5 days. $\Delta\overline{T}_{soil}$ for strategy A decreases by 80.96% and 81.43%, respectively, compared with strategy C. The values for t and SP are 0.50 K, 0.19 K, and 0.09 K, and 0.42 K, 0.16 K, and 0.08 K, respectively.

The downtimes of daily operation of all five types of spiral-sloped energy piles under operation strategy C are taken into consideration as reference points for the duration of the system's operation. The average change in soil temperature within the 6 m × 6 m region centered on the energy piles was then fitted to the data. The average change in soil temperature during intermittent operating conditions can be predicted by fitting the data. Equations (6)–(8) display the respective prediction equations.

Global hybrid continuous prediction curves:

$$T_{(SP,t),1} = 291.69117 - 0.0017SP + 0.01291t + 2.062 \times 10^{-5}SP^2 - 2.124 \times 10^{-4}t^2 \quad (7)$$

where $t$ is the number of operating days.

Global intermittent forecast curves:

$$\begin{aligned} T_{(SP,t),2} &= 291.66842 - 0.00134SP + 0.02048t + 1.594 \times 10^{-5}SP^2 - 6.615 \times 10^{-4}t^2 \\ &+ \lfloor t/7 \rfloor \times 0.0657 \end{aligned} \quad (8)$$

Global forecast curves with adjustments:

$$\begin{aligned} T_{(SP,t)} &= 291.66842 - 0.00134SP + \varepsilon_{t,week}t + 1.594 \times 10^{-5}SP^2 - 6.615 \times 10^{-4}t^2 \\ &+ \lfloor t/7 \rfloor \times \Delta T_{week}\varepsilon_{\Delta T,SP} \end{aligned} \quad (9)$$

where

$$\varepsilon_{t,week} = 0.02048 - 0.0023 \times \lfloor t/7 \rfloor + 1.25 \times 10^4 \times \lfloor t/7 \rfloor^2$$

$$\Delta T_{week} = 0.0113 + 0.0703 \times \lfloor t/7 \rfloor - 0.0062 \times \lfloor t/7 \rfloor^2$$

$$\varepsilon_{\Delta T,SP} = 3 \times 10^{-6} \times SP^3 - 10^{-4} \times SP^2 - 0.0059 \times SP + 1.1399$$

Equations (7)–(9) were used to forecast soil temperature changes in the energy pile with SP = 20 operating at strategy C for 6 weeks. The results of these simulations are compared in Figure 9.

For predicting the average soil temperature over a 6-week period, Equation (8) (Predictive-3) has a very good goodness of fit with a determination coefficient of 0.986. Equation (7) (Predictive-2) has a high goodness of fit only within 2 weeks, and notably, the amount of dispersion increases when it exceeds 2 weeks, indicating that the temperature increase in a single week is not a linear change over time. It is evident that Equation (6) (Predictive-1) still has a high goodness of fit within 3 weeks with a determination coefficient of 0.70 but drops to 0.47 after 4 weeks.

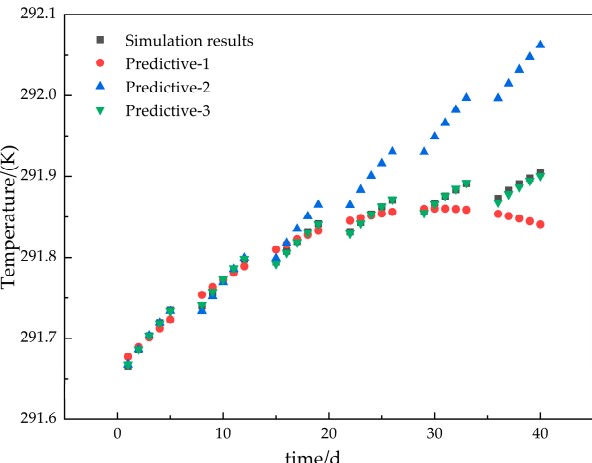

**Figure 9.** Predicted results vs. actual simulation results.

### 3.2. Law of Influence of Helical Pitch on Average Temperature of Various Regions of Energy Piles

Equation (10) yields the dimensionless average temperature ($\theta$) (equivalent thermal resistance R) for every region within the SGHEs-P based on the G-function method [14]. As illustrated in Figure 9, the dimensionless total average temperature ($\theta_{avg}$) of the energy pile is divided into the dimensionless average temperature of fluid ($\theta_{f,avg}$), the dimensionless average temperature of the pile foundation ($\theta_{int,avg}$), and the dimensionless average temperature of soil ($\theta_{ext,avg}$),

$$\theta avg = kghb(\overline{T}f - \overline{T}g)/\Phi f = \theta f, avg + \theta \text{int}, avg + \theta ext, avg = Ravg \tag{10}$$

where *R* is the dimensionless equivalent thermal resistance; subscripts *avg*, *int*, and *ext* stand for the energy pile and soil, pile foundation, and soil.

Figure 10a shows that the soil's equivalent thermal resistance ($R_{ext,avg}$) rises with time but does not with increasing SP. When Fo < 0.1 (t < 7.5 h), the soil temperature gradient ($\theta_{ext,avg}$) is not sensitive to Fo and heat exchange is centered between the spiral pipe and the pile foundation. The exterior wall of the pile foundation will temporarily preserve the starting temperature since the concrete pile foundation is thermally inert. The rise in temperature of the pile foundation's exterior wall causes $\theta_{ext,avg}$ to become more responsive to Fo when 0.1 < Fo < 10. When Fo > 12, the pile foundation's heat transmission capacity saturates, constantly releasing heat into the soil and increasing $\theta_{ext,avg}$'s reactivity to Fo.

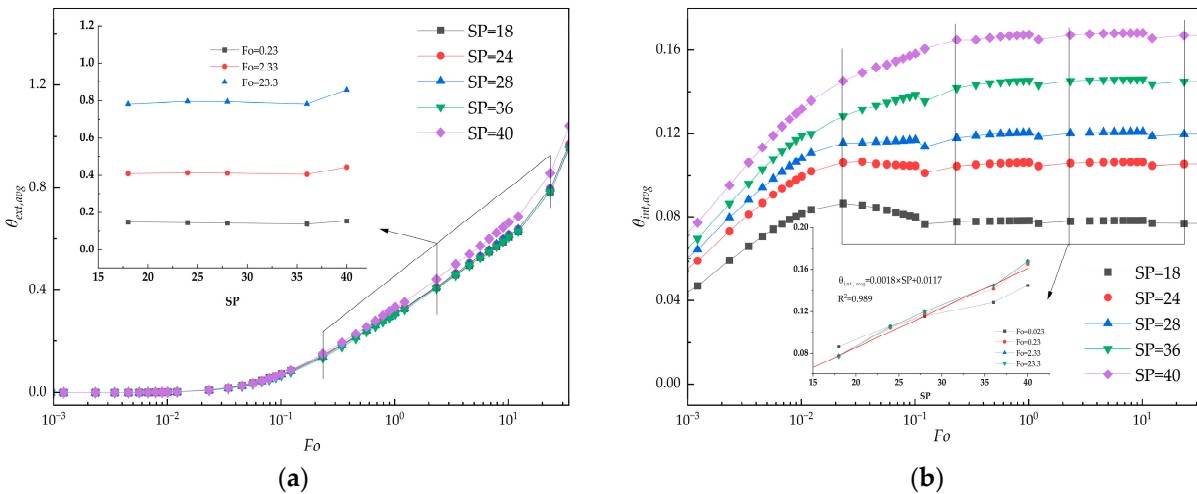

**Figure 10.** *Cont.*

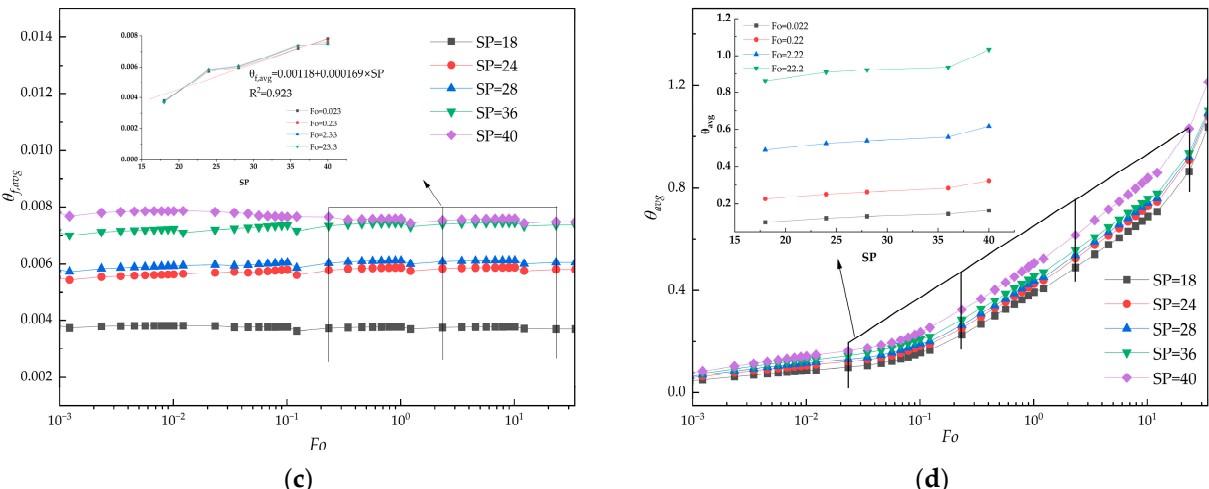

**Figure 10.** Dimensionless distribution of energy piles at temperature (**a**) $\theta_{ext,avg}$, (**b**) $\theta_{int,avg}$, (**c**) $\theta_{f,avg}$, (**d**) $\theta_{avg}$.

Figure 10b shows that the equivalent thermal resistance of the pile ($R_{int,avg}$) increases as *SP* grows. If Fo < 0.1, $\theta_{int,avg}$ increases as Fo increases; when Fo > 0.1, $\theta_{int,avg}$ stays constant, increases linearly when SP increases, and does not change over time. Figure 10c shows that the fluid's equivalent thermal resistance ($R_{f,avg}$) increases linearly with increasing SP, while the fluid's average temperature ($\theta_{f,avg}$) changes by less than ±4% when Fo increases, suggesting less correlation with Fo. Figure 10d illustrates how the energy piles' total equivalent temperature ($\theta_{avg}$) increases as Fo increases, increases with increasing SP at each Fo, and increases in response to SP as Fo increases.

Figure 11 illustrates how the ratios of the energy pile's liquid, external, and internal thermal resistance and its total thermal resistance with *Fo* increasing, as well as how the ratio of liquid thermal resistance and total thermal resistance with t, vary for each of the three operating strategies.

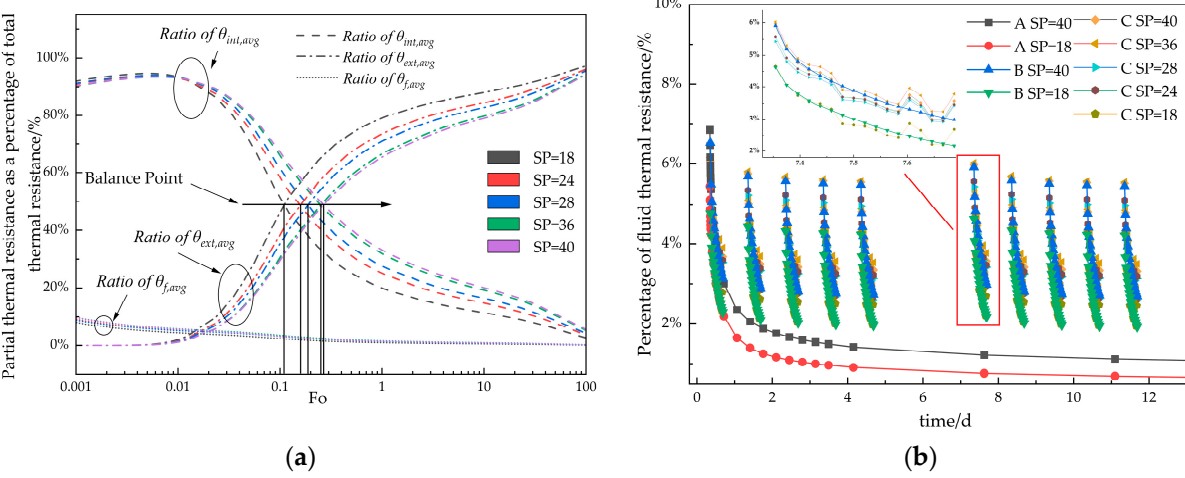

**Figure 11.** The ratios of each component of the energy pile's thermal resistance to total thermal resistance are as follows: (**a**) the ratio of each component's thermal resistance to total thermal resistance; (**b**) the ratio of fluid thermal resistance to total thermal resistance.

Figure 11a shows that the weight of $R_{int,avg}$'s influence on $R_{avg}$ decreases as Fo increases, while the weight of $R_{ext,avg}$'s influence increases. The equilibrium point is reached at 0.1 < Fo < 0.2 (7.5 h < *t* < 15.5 h), and the smaller the SP, the earlier the equilibrium point occurs. It can be seen that for intermittent runs with a 1-day cycle(Fo < 0.2, *t* < 15.5 h), the short-term influence of $R_{f,avg}$ and $R_{int,avg}$ on $R_{avg}$ is significant. (With increasing time, the

influence of $R_{f,avg}$ on $R_{avg}$ eventually diminishes to 2% from up to 20%. The influence of $R_{int,avg}$ on $R_{avg}$ dominates at the initial stage of heat transfer and gradually decreases with increasing Fo to the point where $R_{int,avg}$ equals $R_{ext,avg}$, and then $R_{int,avg}$ gradually decreases until the heat transfer capacity of the pile base temperature is saturated. In Figure 11b, it can be seen that the percentage of $R_{f,avg}$ in $R_{avg}$ is less than 8% in both continuous and intermittent operation, while the percentage of $R_{f,avg}$ in $R_{avg}$ becomes smaller and smaller as the operation time increases, and the percentage of $R_{f,avg}$ in $R_{avg}$ decreases from 6% to 2% in each cycle under the strategy of intermittent operation from the beginning of operation to the end of operation. Therefore, in the intermittent operation strategy, the change in the physical parameters of the turbulent fluid will improve the heat transfer effect of SGHEs-P much more than in the continuous operation strategy, but the effect of the fluid on the heat transfer effect of the energy pile is still at a low level for the whole energy pile.

Therefore, office buildings can successfully reduce the fluid's equivalent thermal resistance and the average temperature at the pile base by implementing a geothermal heat pump system with an intermittent operation strategy for central cooling and reducing the helical pitch. Consequently, this improves the energy pile's overall performance and heat transfer.

### 3.3. Relationship between Spiral Tube Design and Convection Heat Transfer Coefficient

Taking the center point $p_0$ of the normal plane at the middle of the spiral as the origin of the coordinates and establishing a polar coordinate system $(\omega, r, z)$ is shown in Figure 12. The coordinates of $\omega$, $r$, and $z$ in Figure 11 were calculated using Equation (11).

$$(\omega, r, z) = \left( \frac{2\pi rs}{P0} \cos\left(\frac{2\pi z}{P0}\right), \frac{2\pi rs}{P0} \sin\left(\frac{2\pi z}{P0}\right), z \right) \tag{11}$$

where $\omega$ represents the tangential direction of the spiral line and points downstream, $r$ is the radial coordinate, and the $z$ direction is downward along the helix central axis.

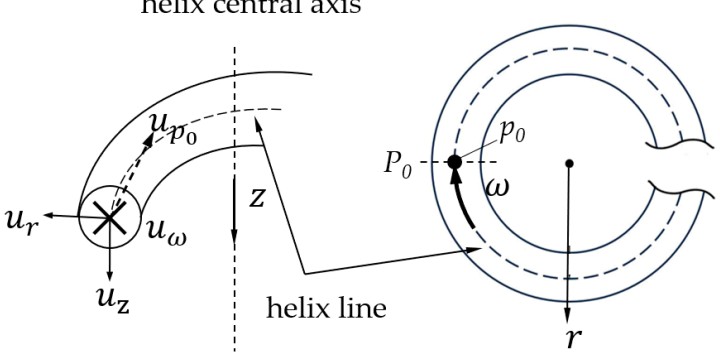

**Figure 12.** Velocity($u_{p0}$) and velocity components at point $p_0$ in the $\omega$, $r$, $z$ directions.

The velocity fields and temperature fields of the normal plane are shown in Figure 13. The velocity field is shown in the upper part, and the temperature field is presented in the lower part of the subfigure.

Figure 13 shows the changes in the secondary circulation in this plane under three conditions: when $R$ and $d$ are kept constant and only SP is changed, when $P$ and $d$ are kept constant and only SR is changed, and when $P$ and $R$ are kept constant and only Sd is changed. It can be observed that the flow rate of the secondary circuit increases with an increase in SP and decreases with an increase in SR and Sd. Additionally, the temperature of the secondary circuit decreases with an increase in SR and *Sd* but increases with an increase in Sd.

Using the controlled variable method, SP, SR, and Sd were adjusted separately to compare the magnitude of the velocity at point p0 under the three independent influences

($u_{p0}$) and decompose the velocity at point $p_0$ into the component velocities in the directions $\omega$, $r$ and $z(u_{p0,\omega}, u_{p0,r}, u_{p0,z})$, as shown in Figures 14 and 15, respectively.

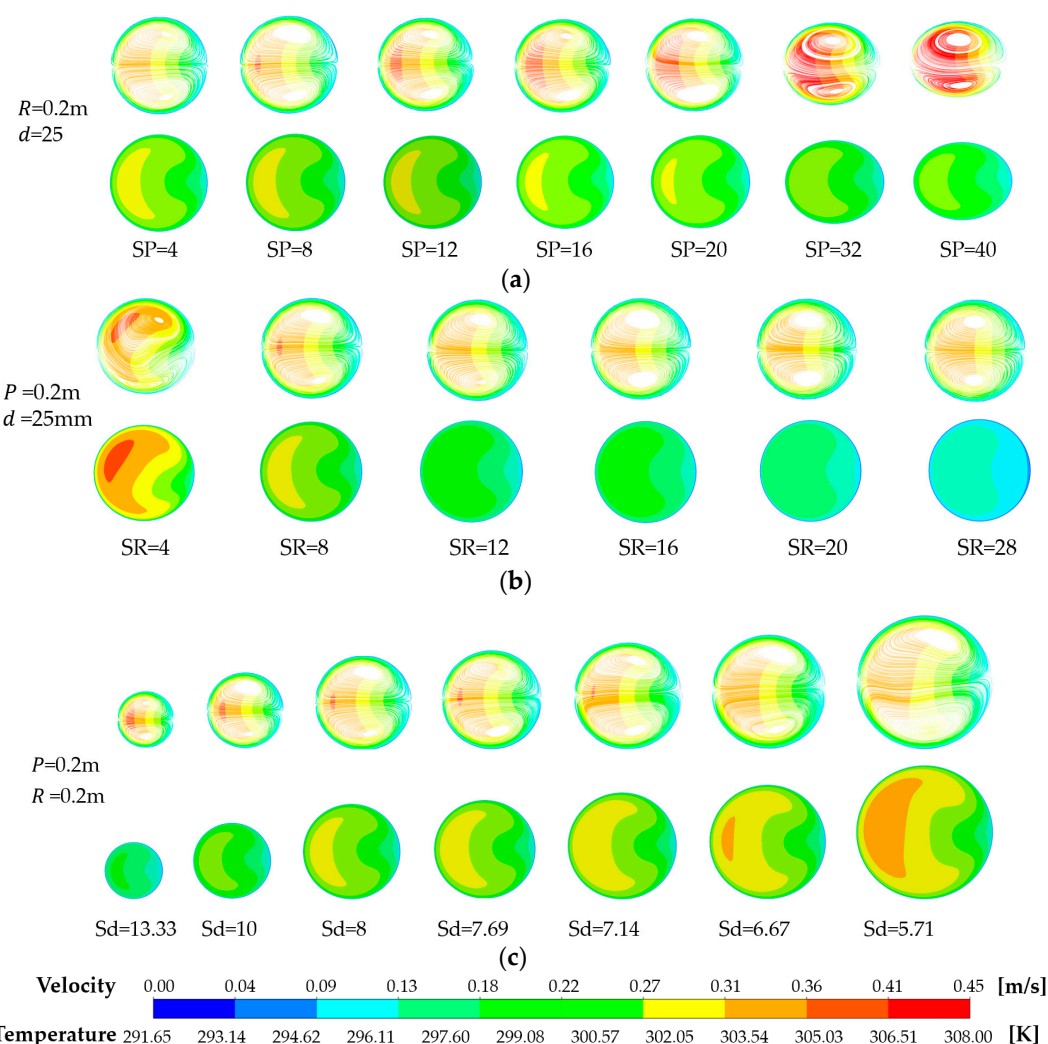

**Figure 13.** Velocity field (upper panel) and temperature field (lower panel) in the normal plane at $1/2P_0$ (secondary circulation): (**a**) keeping $R$ and $d$ constant only changes $SP$; (**b**) keeping $P$ and $d$ constant only changes $SR$; (**c**) keeping $P$ and $R$ constant only changes $Sd$.

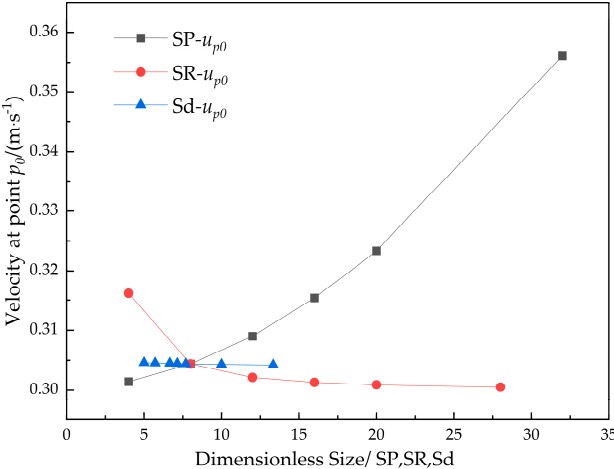

**Figure 14.** Relationship between $u_{p0}$ and pipe dimensions.

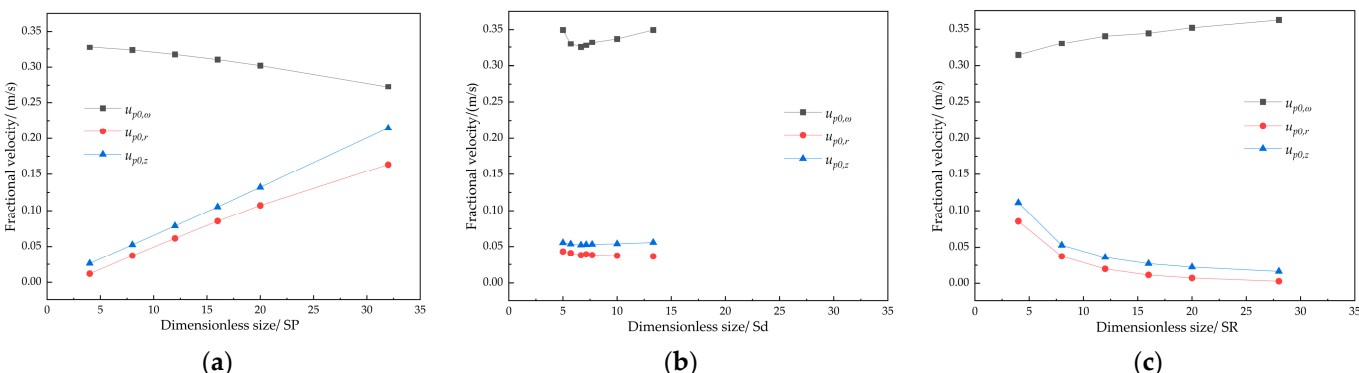

**Figure 15.** Relationship between $u_{p0,i}$ and pipe dimensions: (**a**) Dimension-SP; (**b**) Dimension-Sd; (**c**) Dimension-SP.

It can be seen in Figure 14 that $u_{p0}$ increases by 18.1% with an increase in SP; $u_{p0}$ decreases by 3.24% as SR increases from 4 to 12, then remains stable; an increase in Sd reduces $u_{p0}$ slightly, but only by 0.11%. Figure 15 shows that when SP increases, $u_{p0,\omega}$ decreases by 16.87% and $u_{p0,r}$ and $u_{p0,z}$ increase by 12.7 and 7.2 times; when SR increases, $u_{p0,\omega}$ increases by 15.3%, and $u_{p0,r}$ and $u_{p0,z}$ decrease by 96.7% and 82.5%; and when Sd decreases, there is not much change in $u_{p0,\omega}$, $u_{p0,r}$ and $u_{p0,z}$, less than 5%.

Meanwhile, *SP* and *SR* have significant effects on the flow state of the fluid inside the spiral tube, and the effect of *SP* on the flow state of the fluid inside the spiral tube is greater than that of *SR*, while the effect of *Sd* is very small. When *SP* > 20 and when *SR* < 5, the velocity distributions of $\omega$, $r$, and $z$ of $u_{p0}$ at $p_0$ are similar, and $u_{p0,\omega}$ varies within ±20% of the mean flow velocity across the cross-section. $u_{p0,r}$ and $u_{p0,z}$ are both greater than 0.75 m/s, resulting in $u_{p0}$ being much larger than the average flow velocity in the cross-section, which reduces the flow velocity of the fluid near the wall of the spiral tube and increases the thickness of the boundary layer and, therefore, cannot form a stable secondary cycle.

According to the Dittus–Boelter formula in the cooling case (see Equation (12)), the convective heat transfer coefficient ($\alpha_{aff,st}$) of the inner wall of the smooth circular tube is only related to the tube diameter d, and there is no direct relationship with the length of the circular tube. The ratio of the heat transfer coefficient between the flow-facing inner wall of the spiral tube ($\alpha_{aff,sp}$) and the heat transfer coefficient between the inner wall of the smooth, flow-facing circular tube with the same diameter is referred to as the correction coefficient for the spiral tube ($\varepsilon$); the relationship to Sd, SR, and SP ($\varepsilon_d$, $\varepsilon_{rs}$, $\varepsilon_P$) is shown in Figure 16.

$$Nu = \alpha_{aff} \cdot de/v = 0.023Re^{0.8}Pr^{0.3} \tag{12}$$

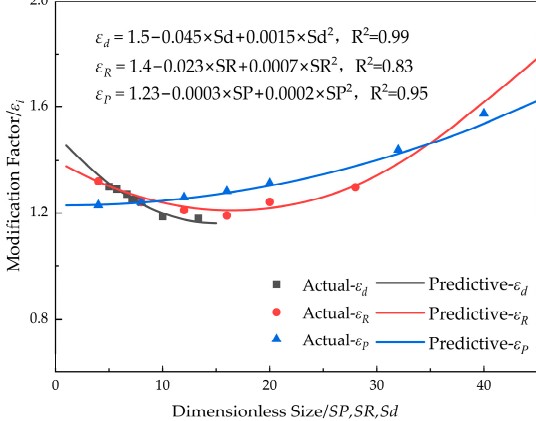

**Figure 16.** Correction coefficients ($\varepsilon_d$, $\varepsilon_{rs}$, $\varepsilon_P$) versus dimensionless design dimensions.

It can be seen that $\varepsilon_P$ increases quadratically with SP as SP increases. As SR increases, there is a quadratic relationship between $\varepsilon_R$ and SR, and there is a minimum ability to improve the effect of convective heat transfer. The correction coefficient $\varepsilon_d$ decreases as *Sd* increases. Among the three groups of independent influences, increasing SP enhances $\alpha_{aff,sp}$, while increasing Sd reduces it. Outside the range of *SR*'s minimum enhancement ability, increasing or decreasing SR has a positive effect on the enhancement effect. The predictive equations for $\varepsilon_d$, $\varepsilon_{rs}$, and $\varepsilon_P$ can be fitted using the software Origin 2024SR1 as Equation (12), Equation (13), and Equation (14), respectively.

$$\varepsilon_d = 1.5 - 0.045 \times Sd + 0.0015 \times Sd^2 \tag{13}$$

$$\varepsilon_{rs} = 1.4 - 0.023 \times SR + 0.0007 \times SR^2 \tag{14}$$

$$\varepsilon_P = 1.23 - 0.0003 \times SP + 0.0002 \times SP^2 \tag{15}$$

The convective heat transfer coefficient in the spiral coil can be computed as follows using the correction coefficients that were previously determined:

$$\alpha_{aff} = \varepsilon \times [(Nu \cdot v)/de] = \left[\varepsilon_p - 0.5 \times (2.4832 - (\varepsilon_d + \varepsilon_{rs}))\right] \times \left[\left(0.023 \cdot v \cdot Re^{0.8} Pr^{0.3}\right)/de\right] \tag{16}$$

## 4. Conclusions

This study used the G-function approach to analyze the heat transfer model of a parallel double helix energy pile at different pitches, and it studied and analyzed the temperature and velocity fields of a spiral pipe at various operating settings. The following conclusions were drawn from the results:

(1) Lowering the helical pitch can significantly raise the SGHEs-P's heat transmission capacity while also improving the pile foundation's horizontal and vertical temperature distribution uniformity.

(2) Over an extended period of time, intermittent operation, as opposed to continuous operation, can achieve a higher and more consistent daily heat transfer and successfully lessen heat damage to the soil that the energy pile system causes.

(3) Under the intermittent operation approach, fluid thermal resistance makes up a clearly larger percentage of the energy pile's overall thermal resistance than it does under the continuous operation method. As a result, under the intermittent operation strategy, the fluid properties will have a greater influence on the energy pile's heat transfer effect.

(4) The fluid velocity in the tube's core spirals faster as the spiral pitch rises, thickening the boundary layer. When the dimensionless pitch (SP) exceeds 32, a stable secondary circulation in the spiral cross-section would not been observed obviously, which hinders heat exchange and vice versa. It was also discovered that variations in spiral tube diameter have no appreciable impact on heat transfer.

(5) In order to more precisely calculate the convective heat transfer coefficient in the spiral tube, correction factors for spiral pitch, spiral diameter, and spiral tube diameter were determined based on the simulation findings. A convective heat transfer coefficient calculation method was presented in relation to three correction variables. There is a significant link between the correction coefficient and SR and SP. For $\varepsilon_{rs}$, a local minimum was seen, with SR = 16 resulting in the least value. Consequently, one can improve $\varepsilon_{rs}$ by modifying the surface roughness (SR) in an upward or downward direction.

**Author Contributions:** Writing—original draft preparation, P.W. and K.L.; writing—review and editing, P.W. and K.L.; software—P.W., C.Y. and K.L.; resources, Q.H. All authors have read and agreed to the published version of the manuscript.

**Funding:** This research was funded by the "National Natural Science Foundation of China (NSFC)", grant number 42202321.

**Institutional Review Board Statement:** Not applicable.

**Informed Consent Statement:** Not applicable.

**Data Availability Statement:** The data presented in this study are available on request from the corresponding author.

**Conflicts of Interest:** The authors declare no conflict of interest.

## Nomenclature

| | |
|---|---|
| $T$ | Temperature, K |
| $P$ | Helical pitch, m |
| $r_s$ | Helix radius, m |
| $d$ | Diameter of spiral tube, m |
| $de$ | Characteristic length, m |
| $h$ | Height, m |
| $L$ | Length, m |
| $\rho$ | Densities, kg/m$^3$ |
| $c_p$ | Specific heat capacity, J/kg·K |
| $k$ | Thermal conductivity, W/m·K |
| $q$ | Heat flux, J/m$^2$ |
| $\alpha_{aff}$ | Convective heat transfer coefficient, W/m$^2$·K |
| $u$ | Flow velocity, m/s |
| $\Phi$ | Heat transmission, J |
| $R$ | Dimensionless equivalent thermal resistance |
| $\theta$ | Dimensionless temperature |
| $Re$ | Reynolds number |
| $Pr$ | Prandtl number |
| $Fo$ | Fourier number |
| $Nu$ | Nussle number |
| $C$ | Inaccuracy |
| $Q$ | Heat exchange, W |
| **Greek alphabet** | |
| $\tau$ | Time, s |
| $\mu$ | Dynamic viscosity, Pa·s |
| $\nu$ | Kinematic viscosity, m$^2$/s |
| $\varepsilon$ | Correction factor |
| $\gamma$ | Volume ratio |
| **Suffix** | |
| $st$ | Tubes |
| $sp$ | Spiral tube |
| $f$ | Fluid |
| $g$ | Soil |
| $p$ | Pile |
| $b$ | Pile wall |
| $int$ | Internal |
| $ext$ | External |
| $avg$ | Average |
| $tube$ | Tube wall |

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
