# Peer review of "Effect of Different Control Strategies on the Heat Transfer Mechanism of Helical Energy Piles"

_applsci, doi:10.3390/app14072836_

Round 1

Reviewer 1 Report

Comments and Suggestions for Authors

The manuscript discusses the effect of different spiral pitches on heat transfer and soil temperature for a Spiral Ground Heat Exchanger Systems by means of numerical model developed in ANSYS Fluent. The manuscript is well organized and the topic is clear. Some formatting errors are present and images need to be more clear and understandable. A discussion on the effect of heat transfer of piles in subsoil concerning the groundwater flow issue is required.

Here further specific comments.

Many formatting error are present in the manuscript. Please correct.

In introduction a general section discussing the role of Borehole Heat Exchangers consisting of single or double U-pipe and how they are modeled is needed. Please read and refer to these manuscript:

-        Antelmi et al. A new type in TRNSYS 18 for simulation of Borehole Heat Exchangers affected by different groundwater flow velocities, Energies 2023, 16, 1288; https://doi.org/10.3390/en16031288,

-        Alberti et al., Borehole Heat Exchangers in aquifers: Simulation of the grout material impact, Rendiconti Online Societa Geologica Italiana, https://doi.org/10.3301/ROL.2016.145

-        Casasso, Sethu, Efficiency of closed loop geothermal heat pumps: A sensitivity analysis, Renewable Energy, https://doi.org/10.1016/j.renene.2013.08.019

The last part of introduction has to deeply highlight the target of the manuscript. How this manuscript is improving this sector? Which is the innovation of your discussion?

Please correct the format of table 1 and 2

Figure 3 is not clear, it is too little.

Please describe in detail all the boundary conditions set in the model and also the modeling grid mesh.

Which is the effect of the groundwater flow on the heat transfer for the two case simulated? You write of a flow equal to 0.3 m/s; please discuss in detail. The importance of groundwater flow in heat transfer effects is very important according to the previous suggested manuscripts.

The discussion and conclusion are adequate but the innovation of this work is missing. Please discuss how your study improves the geothermal literature overview.

Comments on the Quality of English Language

Please be careful to grammar erros; a further check on language is needed

Reviewer 2 Report

Comments and Suggestions for Authors

The submitted manuscript solves the interesting issue of pumping geothermal energy, respectively it solves the transfer of heat from ground heat exchangers. The problems arising in the solution of these types of exchangers are connected to the properties of the environment and the parameters of the heat exchangers. Many studies have confirmed the impossibility of using ground heat continuously, but heat regeneration around the exchangers is necessary.

The presented research showed other ways of solving more efficiently using earth's heat using earth's heat exchangers. Nevertheless, I have a few formal and professional comments.

1) When describing mathematical relationships, several quantities were used, which are not fully or not at all described even in the text. It is necessary to add to the manuscript Nomenclature.

2) In the introduction, there are non-functional references to the used literature, respectively, only error messages associated with changing the references to the literature.

3) The authors used the value of the dimensionless criterion Fo in the text, but it is not clear where these values are obtained. It is necessary to improve the description in the text.

4) The designation "temperature gradient (θ)" and "thermal resistance (R)" is questionable. Because in thermal engineering calculations, "θ" is used as dimensionless temperature in calculations to determine the temperature using "Fo" and it contradicts the actual designation of the temperature gradient "grad t". That is also why in this context it is questionable to equate "temperature gradient (θ)" and "thermal resistance (R)". Thermal resistance is related to the physical properties of the material and its thickness, so it is unclear what the authors meant by this.

5)In the text, the authors compare some quantities and the statements are more general. It is necessary to specify the increase or decrease at least in percentages instead of phrases using expressions such as in the text under Fig.13. "being much larger than", etc.

I agree with the statement in "Conclusion" about the necessity of regeneration or discontinuous operation of such earth exchangers. As pointed out in this manuscript, the answer can be by changing the design parameters and the way of operating such types of exchangers.

Round 2

Reviewer 1 Report

Comments and Suggestions for Authors

The suggestions previously provided are now applied and the quality of the manuscript is improved.

Please refer to the format defined by authors guideline for text and figures and be careful with grammar errors.

Comments on the Quality of English Language

Some type errors need to be correct

Reviewer 2 Report

Comments and Suggestions for Authors

The authors of the manuscript have revised the manuscript according to the comments and the improvement of the manuscript can be seen. The treated topic is very interesting and brings valuable knowledge for further research. I have no serious comments on the revised manuscript. Nevertheless, I think that in the case of the use of abbreviations and the number of mathematical relationships used, it is necessary to add Nomenclature to the manuscript. I recommend checking the citations in the entire text. On line 66, there is the same citation twice but marked differently.
